# Addressing Tobacco Use in Underserved Communities Outside of Primary Care: The Need to Tailor Tobacco Cessation Training for Community Health Workers

**DOI:** 10.3390/ijerph20085574

**Published:** 2023-04-19

**Authors:** Marcia M. Tan, Shariwa Oke, Daryn Ellison, Clarissa Huard, Anna Veluz-Wilkins

**Affiliations:** 1Department of Public Health Sciences, University of Chicago, 5841 S Maryland Ave, Chicago, IL 60637, USA; mmtan@bsd.uchicago.edu (M.M.T.); t-9veluz@uchicago.edu (A.V.-W.); 2Department of Preventive Medicine, Feinberg School of Medicine, Northwestern University, 680 N Lake Shore Dr, Chicago, IL 60611, USA

**Keywords:** tobacco cessation, tobacco cessation interventions, community health workers, health disparities

## Abstract

Individuals from communities with a low socioeconomic status have the highest rates of tobacco use but are less likely to receive assistance with quitting. Community health workers (CHWs) are well-positioned to engage these communities; however, CHWs face barriers in receiving relevant tobacco cessation training. The objective of this study was to conduct a mixed methods needs assessment to describe tobacco practices and the desire for training among CHWs. After incorporating CHW feedback, we developed a needs assessment survey to understand knowledge, practices, and attitudes about tobacco cessation in Chicago, IL. CHWs (N = 23) recruited from local community-based organizations completed the survey online or in-person. We then conducted a focus group with CHWs (N = 6) to expand upon the survey and used the Framework Method to analyze the qualitative data. CHWs reported that their clients had low incomes, low literacy levels, and high smoking rates (e.g., “99%” of patients). About 73.3% reported discussing tobacco use during visits, but fewer reported that they had provided cessation advice (43%) or intervened directly (9%). CHWs described high variability in their work environments (e.g., location, duration, content of visits, etc.) and greater continuity of care. CHWs discussed that existing training on how to conduct tobacco interventions is ineffective, because of its stand-alone design. Our findings illustrate how CHWs adapt to their clients’ needs, and that the currently available “gold-standard” cessation curricula are incompatible with the training needs and flexible care delivery model of CHWs. A curriculum tailored to the CHW experience is needed to maximize the strengths of the CHW care model by training CHWs to adaptively intervene regarding tobacco use in their highly burdened patients.

## 1. Introduction

Although rates of cigarette smoking have declined in the US, low socioeconomic status (SES) populations smoke at a higher rate than the national average (24% compared to 15%) [1]. Brief advice for smoking cessation given by a physician or nurse increases the quit rate compared to situations in which no advice is given [2,3]. As per the US Public Health Service recommendations, brief interventions should be used by healthcare providers to address tobacco use, particularly in the clinic setting [4]. However, these brief interventions have not been administered adequately to adults with low SES and miss a large segment of those who smoke. During primary care visits in community settings, healthcare providers (primarily physicians) ask patients about their tobacco use; yet, of those identified as currently smoking, only about 21% receive counseling to quit [5]. Those with an unknown health insurance status are less likely to receive counseling compared to those with a known insurance status [5]. According to previous research, patients with a low SES and those from marginalized groups are even less likely to receive assistance with quitting [6,7]. For example, in a population-based sample, adults with high levels of socioeconomic disadvantage were 41% less likely to report receiving cessation support from a healthcare provider compared to those with a low level of disadvantage [6]. Moreover, approximately 40% of adults with low incomes do not attend primary care visits annually [5,6]. Given the limitations of the availability of brief interventions, adults with a low SES need smoking cessation interventions that reach them outside of traditional primary care clinic visits. This group can benefit from evidence-based tobacco treatment modalities, such as counseling [4], and the Clinical Practice Guidelines for Treating Tobacco Use and Dependence [4] call for research on the effectiveness of evidence-based treatment in community-based settings. Community health workers (CHWs) have contact with individuals who smoke in nontraditional settings; therefore, they are an integral part of delivering treatment directly to low SES communities with the highest tobacco-related burden.

CHWs are frontline public health workers who are trusted members of the communities in which they serve. They are known by several different titles, including lay health advisors, community health advisors, and *promotores de salud* (in Latinx, Spanish-speaking communities). Because CHWs are members of the community, they are more likely to influence health decisions and communicate health information than general healthcare providers found in traditional healthcare systems [8]. In the US, CHWs have been involved for decades in broad prevention efforts, such as screening, healthy lifestyle promotion [9,10], and disease and medication management [11,12,13], and the role of CHWs was officially recognized as part of the healthcare workforce under the Affordable Care Act in 2010 [14]. Trained CHWs have been shown to play an increasing role in tobacco cessation programs in the community [15,16]. Previous research has shown that incorporating CHWs as interventionists in tobacco cessation programs has had some positive effects on quit rates [17]. For example, Woodruff et al. (2002) reported an increase in biochemically confirmed past-week abstinence rates for participants (*n* = 132) enrolled in a tobacco cessation program led by a community health advisor compared to participants (*n* = 150) who were referred to a state quitline (21% and 11%, respectively). Community health advisors in this study were trained on culturally appropriate smoking cessation intervention content for Latino individuals who smoke that could be delivered to participants in their homes. Of those who were trained, approximately 89% delivered at least one intervention to their patients [18]. Previous research has shown not only that CHWs can be effectively trained in tobacco cessation delivery [17,19,20,21], but also that CHWs are confident in delivering interventions to their patients after receiving training [21].

Despite the effectiveness of CHW training on smoking cessation outcomes, studies have shown that CHWs still lack core tobacco knowledge [19,22]. This lack of knowledge may be due to the inaccessibility of training for CHWs [22]. Firstly, organizations have adopted the CHW care model to varying degrees; therefore, the type of training received by CHWs is largely dependent upon what their employers offer or reimburse [23]. For example, the cost of in-person facilitator training sessions for evidence-based group or individual tobacco treatment programs ranges from $150 to $400, and the training usually lasts a full day [24,25]. Secondly, existing training for conducting cessation interventions may not be suitable for CHWs, as they do not take into account the unique position of CHWs as healthcare promoters in their own communities. For example, research has shown that lay health workers, including CHWs, have concerns about their social positions with their clients when conducting interventions [26]. Research also has shown that formal, consistent training is useful for increasing self-efficacy in delivering interventions [20,27].

Research suggests that training on tobacco cessation for CHWs needs to be both (a) accessible and (b) relevant to CHWs. However, specific ways to target training sessions for CHWs are unclear. The scientific community has called on research funders and institutions to make community engagement mandatory in clinical and translational research in order to improve the quality of the science, the relevance of the work, and the translation to practice [28]. Therefore, we aimed to take a community-engaged approach to understand the needs of tobacco cessation training among CHWs. The objective of the current study was to conduct a mixed methods needs assessment to (a) describe current tobacco practices and desires for training among CHWs and (b) to understand the CHW model of care and how a training session might best be tailored to CHWs to help their patients who smoke.

## 2. Materials and Methods

### 2.1. Participants

Community health workers and allied health professionals were recruited from local community-based organizations (CBOs) and email listservs to complete a needs assessment survey online or in-person. Participants with titles of “community health worker,” “community health educator,” or “promotora de salud,” who were employed or volunteering at CBOs and/or healthcare organizations were included in the current study.

### 2.2. Procedure

To understand current efforts to treat tobacco use among patients, we used an explanatory sequential mixed methods design [29] consisting of a quantitative survey followed by a focus group to explain the initial results. We previously partnered with tobacco control specialists at the American Lung Association and CHW-employed CBOs to conduct a needs assessment in Chicago, IL. Chicago, the third-most populous city in the US, has disproportionately high rates of cigarette smoking among racial/ethnic minority adults (25% among African Americans compared to 13% of African Americans in the general population) and among those with lower incomes (26% compared to 20% in the general population) [30]. Furthermore, Chicago has a poverty rate of 17% [31], and among adults living below the poverty line, only about 18% reported that they had a primary care provider and 23% reported that they had received a routine checkup in the previous year [30].

*Data collection.* Quantitative data were extracted from a survey that was developed to understand knowledge, practices, and attitudes about tobacco cessation among CHWs and nonphysician lung health professionals [22]. Local CHWs, who were employed at a CBO affiliated with a large, urban, healthcare system, had provided feedback on the development of the survey questions to ensure its relevance to CHWs and their patients. Qualitative data were collected via a 60 min, in-person, semi-structured focus group to expand upon our quantitative data. Focus group participants were CHWs who had been recruited via email outreach from CBOs that serve the most vulnerable neighborhoods in the city, and the focus group was held at their offices to increase access and convenience for the participants. We informed participants that their participation or lack of participation did not affect their employment status at their workplace. Participants completed a demographic questionnaire at the start of the session and received a gift card for participating in the session. The focus group was moderated by a doctoral-level clinical health psychologist (MMT) and a master’s-level health psychologist (AVW). To guide the discussion, we created a focus group guide using results from the needs assessment survey to ensure that we were expanding upon the identified needs of the CHWs. The study was approved by the Institutional Review Board.

*Qualitative coding procedure.* Qualitative data were transcribed verbatim by a member of the research team and coded using the Framework Method, according to the principles outlined in Gale et al. (2013) [32]. Two coders, a Master’s-level health psychologist (AVW) and a Master’s-level trained research assistant (SO), independently reviewed the focus group transcript to generate preliminary codes relevant to the research questions. AVW and SO then collaboratively refined the code definitions, including primary (i.e., major topics explored) and secondary codes (i.e., recurrent themes within these major topics) to create a working analytical framework and a coding dictionary. The transcripts were independently reviewed again using the coding dictionary to ensure accuracy and consistency. The coders met regularly to refine and clarify codes and definitions, and a consensus for an analytical framework was reached.

### 2.3. Measures and Analyses

*Quantitative data.* The 46-item needs assessment survey, which consisted of multiple choice and checkbox questions, assessed information on job and client characteristics, tobacco cessation practices, desired skills and training sessions, and a 10-item knowledge questionnaire about tobacco use and cessation in the U.S. 

*Focus group guide.* We developed a semi-structured guide that was used to lead the discussion during the focus group. Participants were asked questions such as “To get started, can you tell me a little bit about your interactions with your clients/patients?” and “Do you feel confident in addressing tobacco use/dependence with your clients/patients? Why or why not?” We used descriptive statistics (frequencies and percentages) to summarize the quantitative results.

*Qualitative data.* Taking a broadly inductive approach, we then conducted thematic analyses of the qualitative data using the Framework Method, which is commonly used for the analysis of semi-structured focus group transcripts [32]. After refinement of the codebook and definitions, the transcripts were independently coded and compared. The main themes were extracted from codes that were applied frequently and were used to describe current tobacco cessation practices and needs to consider when tailoring a training session for CHWs. The study team met frequently to discuss the summarized data and reach a consensus for interpretation.

The reporting of these qualitative methods and results conformed with the COREQ (COnsolidated criteria for REporting Qualitative research) guidelines [33]. We followed these guidelines (see Appendix A) to ensure the rigor and trustworthiness of our study design and results, according to best practices for qualitative methods [34].

## 3. Results

We collected quantitative needs assessment survey data (N = 23) and qualitative focus group data (N_Group Participants_ = 6) from CHWs from the Chicago, IL area. Quantitative results of the sample characteristics and tobacco services and practices are presented in Table A1. Major themes identified during the qualitative analyses that may be particularly pertinent to CHW tobacco training needs are presented in Table A2 and are summarized below. Qualitative and quantitative results are discussed thematically below.

### 3.1. Type of Clients

Qualitative focus group data and quantitative needs assessment data indicated that CHWs serve a unique and often higher-risk clientele. The survey results indicated that CHWs serve a racially and ethnically diverse patient population, with the majority of CHW serving Black and Hispanic/Latinx patients (62% and 55%, respectively). About 65% also reported that they worked with sexual and gender minorities. Participants in the focus group discussed that many of their patients have a “*low income*”, “*low health literacy*”, and exhibit high smoking rates. Focus group participants almost universally agreed that “*over 90% of* [their patients] *smoke*”. One participant even estimated that “*99%*” of their caseload were active smoking. During the focus groups, CHWs spoke about working with patients that lived in affordable housing where smoking rates are much higher than the national average [35], and 40.3% of survey respondents indicated that they worked with patients who had experienced homelessness.

Another key characteristic of this client population is that they are primarily referred to CHWs because of specific, typically tobacco-related, health conditions. According to the needs assessment survey, 72.4% of CHWs worked with clientele with chronic illnesses such as asthma, diabetes, and cancer; 56.9% worked with those with mental health disorders; and 40.3% had clients with substance use disorders.

### 3.2. Existing Tobacco Services Provided and Available Cessation Training Programs

Although the survey data indicate that about 73.3% of CHWs reported that they had discussed tobacco use during visits, qualitative data demonstrated that CHWs do not have a standard approach when it comes to smoking cessation, e.g., “*…we don’t have like a set protocol on how to convince somebody to quit smoking, we just sort of try to use whatever works for that person*”. In fact, when surveyed about the level to which they address tobacco use/nicotine dependence only 35% reported the provision of cessation advice and only 9% had intervened directly. This could be because not all CHWs had received smoking cessation training. The survey results showed that only 22% of CHWs had access to training at their facility. When asked if they would pay for cessation training, several CHWs immediately said, “*No!*”. One explained further, “*We [CHW] each do about seventeen things—that our pay still doesn’t meet. And then you ask me to pay for some more classes? I’mma co-self learn, and I’mma share these tools with all of us…and we’ll just share the information cause it’s not [a strain on] our budget… I say, ‘hey, Siri!’ and that’s it, that’s it… we got it [requested health information]*”.

One CHW mentioned that she had completed a standard smoking cessation training and attempted to apply the concepts with her clients “*…they thought it was really great, they thought it was very inspiring, very encouraging, but they just…again they still had all of these surrounding influences and factors that just wouldn’t allow them to completely detach from that behavior*”. Another CHW mentioned the use of teaching material from the Courage to Quit^®^ program but without much impact: “*…people will come and complete the educations sessions, but they still smokin*”. A few CHWs stated that they regularly refer patients to ongoing Courage to Quit^®^ groups, but they expressed doubt that their patients could realistically attend, and instead, wanted information that they could easily and quickly integrate into health education they are already providing to patients during visits: “*…if you say, ‘we have a [quit smoking] class once a week’, that’s still what [providers] do already. It’s like nobody’s gonna do that because people don’t double plan their day to go to a class. You gotta get [cessation assistance] in where you can fit it in and for me, if I had the wherewithal to give them that impactful piece that wouldn’t take me more than like 10 min to explain, you know, why wouldn’t I do it then?*”; *“… [quit smoking information] would need to be a piece that can be interwoven into anybody’s [CHW] protocol they have already. So if [smoking] comes up, add this on as part of what you’re doing, and not try to make it a separate class because people ain’t got time for that*”.

### 3.3. CHW Model of Healthcare Delivery

Qualitative focus group data showed that the CHW model of care reflects the goal of reducing health disparities in underserved communities and is distinctive in terms of its flexibility/variability, continuity of care, and relationships with patients, compared to clinic- and/or hospital-based care. Firstly, CHWs are given flexibility in their work environments with regards to the location, duration, and content of their visits compared to those in traditional healthcare settings. They do not usually operate from offices, but rather, go to patients’ homes, or “*wherever [the patients] feel comfortable*”. Beyond home visits, the spaces identified ranged from “*churches*” to “*community centers*” to “*local libraries*” to “*restaurants*”. The duration and timing of these visits also varies, as noted by one focus group participant, “*It could be once a month; it could be every week*”. Variability was most evident in the content and purpose of their visits. CHWs are assigned to visit patients based on specific presenting health conditions, ranging from asthma to breast cancer. They also perform a variety of tasks during their visits, similar to what occurs in clinic-based care, e.g., “*We do data collection… we have a tool that we will* [use to] *get the data…some background information about their connection to doctors, some of their habits, and things of that nature*”. CHWs often have the added advantage of being able to observe their patients’ home environments: “*at that* [first visit] *we do like an overview as far as assessing the environment. But then at our next visit comes a more detailed look*”.

Secondly, CHWs described having more continuity (e.g., weekly and/or monthly visits) built into their model of care than primary care providers, as evidenced by this focus group quote: “*It’s always a continuation process, it never stops*”. CHW also leverage continuity and are careful to scaffold information provided to their patients to encourage learning. As stated by one CHW, they “*…do it in steps… they’re not bombarded all at once with everything, it’s in, like, pieces*”.

Finally, the continuity of visits and flexibility in their meetings enable CHWs to build relationships with their clients that primary care providers are often not afforded. More specifically, while their clients still see them as healthcare providers, CHWs mentioned in focus groups that clients also view them as a “*mother*”, “*sister*”, “*friend*”, and “*equal*”. There is an opportunity to capitalize on these unique characteristics of the CHW model of care by tailoring existing evidence-based nicotine treatment and tobacco-related disease management approaches to address the health equity and healthcare access needs that continue to disadvantage these communities.

## 4. Discussion

Individuals from communities with low SES suffer from persistently high rates of tobacco use and exposure, and targeted interventions are necessary to provide them with adequate smoking cessation treatment. CHWs work to deliberately eliminate barriers that these communities may face in receiving healthcare services and treatments. The results of our mixed methods needs assessment illustrate that CHWs may greatly benefit from smoking cessation training sessions that are better tailored to the way they provide health education to communities, and that the tailored training curriculum should highlight the specific needs of more vulnerable populations who are most likely to engage with CHWs.

One overarching theme of the current study was that the CHW healthcare model is designed specifically to improve access to healthcare and achieve health equity. Therefore, CHWs are flexible not only in their care strategies but also in their interactions with their patients. Existing training sessions for conducting cessation interventions may not be suitable for CHWs, as they do not take into account the unique position of CHWs as healthcare promoters in their own communities. For example, a previous study qualitatively examined the attitudes of lay health influencers (i.e., lay health advisors or educators, CHWs, and promotoras; N = 141) about conducting brief interventions for smoking [26]. The following themes were reported: (1) concern about the impact promoting smoking cessation has on their social relationships, (2) preference for material resources to reduce social tension when interacting with patients, and (3) desire for a “community of practice” among fellow lay health influencers in order to maintain the sustainability of the program. Variables specific to the CHW experience, such as their social positions in their communities, should be considered when designing or tailoring training sessions.

Given the nature of CHW’s interactions with their patients, the way in which tobacco cessation support is delivered may need to be different than standard approaches to tobacco cessation interventions. Materials and methods presented in a standard smoking cessation training session are often designed to be given in a stand-alone form. Previous research has shown that underserved adults who smoke may benefit from manualized, evidence-based smoking cessation interventions [36], especially if they are easily accessible and offered in community settings [37]. Still, completion rates were shown to be low for this population, and results suggested that the likelihood of quitting smoking was higher if individuals completed a longer, more intensive program or were more ready to quit [37]. Notably, we found that CHWs rarely served patients to primarily talk about smoking cessation. CHWs are requesting the integration of cessation education support as part of management of chronic illness, as chronic illness management is often noted as the priority for a CHW visit or for their patients. In addition, repeated, systematic training sessions for CHWs may be beneficial [18,38], especially as CHWs are often involved in more frequent, ongoing care of their patients. CHWs have previously reported having increased self-efficacy when conducting smoking cessation interventions as a result of formal cessation training [26]. Studies also suggest that continuous training sessions may be appropriate for lay health influencers to sustain their intervention delivery over time [20,27].

Our data identify these key areas of cessation support that seem most appropriate to amplify and directly address in CHW-specific tobacco cessation curricula. Existing evidence-based training sessions do not cater for the training needs of CHWs (e.g., often costly and time-consuming all-day or multiday workshops; do not include enough information on managing nicotine dependence with comorbidities), and none have been created to maximize the CHW care model to adequately address the specific needs of the populations served by CHWs. Tobacco cessation trainings need to be bolstered—from both content and format perspectives—and made available to CHWs in order to meet their needs, especially given the integral roles played by CHWs to address disparities within the US healthcare system.

### Limitations

While these results outline new information about cessation support and training needs of CHWs, our study has a few limitations. Like all qualitative data, our findings must not be misrepresented as generalizable to all CHWs and should be understood as reflecting only the practices and needs of those serving within Chicago and, more specifically, the communities on the south and west sides of the city, as these are the primary catchment areas served by those included in our study. Additionally, these data do not include CHWs serving Spanish-speaking populations and, therefore, cannot be used to inform cessation needs among that important subpopulation of patients. We intend to conduct more research in the future to address these gaps.

## 5. Conclusions

Community health workers are members of the community who work alongside healthcare systems to provide basic health and medical care with the community and are well-positioned to engage members of low SES communities in places in which they live and work, increasing access to quality healthcare and addressing associated health inequities. Current “gold-standard” cessation training curricula have not been accessible to or successfully adopted by CHWs, likely because current training sessions may be incompatible with the flexible approach that CHWs take to care delivery. Tailored tobacco cessation curricula should be developed and made available to teach CHWs how to adaptively intervene on tobacco use in a way that maximizes the strengths of their model of care, i.e., flexibility and the ability to maintain follow-up. Even so, CHWs face barriers in receiving proper training on tobacco-related health disparities that affect their communities and could benefit from training tailored to address these disparities and capitalize on the unique features of the CHW care model.

## Data Availability

This study was not formally registered. This was an investor initiated mixed methods study; therefore, the analysis plan was not formally registered. De-identified data from this study are not available in a public archive. De-identified data from this study will be made available (as allowable according to institutional IRB standards) by emailing the corresponding author. The analytic code used to conduct the analyses presented in this study is not available in a public archive. It may be available by emailing the corresponding author. Materials used to conduct the study are not publicly available but can be made available by emailing the corresponding author.

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
