# Peer review of "Addressing Tobacco Use in Underserved Communities Outside of Primary Care: The Need to Tailor Tobacco Cessation Training for Community Health Workers"

_ijerph, 2023, doi:10.3390/ijerph20085574_

Round 1
Reviewer 1 Report
It is a study based on a quantitative and qualitative survey that aims to describe current practices against smoking and perceptions of training among CHW and secondly, how to understand a model that helps their patients quit smoking.
Well-described methods. Well described design. Definition of subjects and tools and well-developed analysis.
In my opinion, and despite the fact that there is a table in an annex, authors should make an effort to present their main results in the form of a table – or graph – for a better and faster understanding by the reader.
In the discussion section, the limitations of the study should be included.
Author Response
We would like to thank all the reviewers for their comments and their time reviewing this manuscript. Responses to the reviewers are found below in bold. Edits to the manuscript have been tracked changed for ease of reviewing.
Reviewer #1
It is a study based on a quantitative and qualitative survey that aims to describe current practices against smoking and perceptions of training among CHW and secondly, how to understand a model that helps their patients quit smoking.
Well-described methods. Well described design. Definition of subjects and tools and well-developed analysis.
- Thank you for your comment.
- In my opinion, and despite the fact that there is a table in an annex, authors should make an effort to present their main results in the form of a table – or graph – for a better and faster understanding by the reader.
- Thank you for the suggestion. We have included a table in the Appendix (Table A2) to present the qualitative results. We also have referenced the table in the Results section (line 179).
- In the discussion section, the limitations of the study should be included.
- We have labeled the Limitations paragraph in the Discussion section (line 315).
Reviewer 2 Report
A well written article. My comments as attached.

Author Response
We would like to thank all the reviewers for their comments and their time reviewing this manuscript. Responses to the reviewers are found below in bold. Edits to the manuscript have been tracked changed for ease of reviewing.
Reviewer #2
- A little more evidence on underserved communities would be good to be included in the introduction.
- We have included more information on underserved communities in the Introduction section (lines 42-48)
- Line 53: Community health workers can be abbreviated to CHW
- We have updated this in the manuscript (line 58).
- Suggest explaining about why Chicago was chosen as the study site.
- We have included a description of the city of Chicago to explain the relevance of Chicago as the study site (lines 115-121).
- Data on healthcare utilization and the SES of the communities in Chicago would be useful for international readers.
- We have included information on healthcare utilization and SES specific to Chicago (lines 120-121).
- Authors may want to consider to cite the following article if it is related:
- Lourdes, T. G. R., Rodzlan Hasani, W. S., Mohd Yusoff, M. F., Abd Hamid, H. A., Mat Rifin, H., Ismail, H., Saminathan, T. A., Yn, J. L. M., Ab Majid, N. L., Riyadzi, M. R., Ahmad, A., & Ramly, R. (2021). Training is an Important Factor for Community Health Workers in Performing KOSPEN Health Screening Activities in Malaysia: Community Health Workers (KOSPEN) 2016. International Journal of Public Health Research, 11(2). Retrieved from https://spaj.ukm.my/ijphr/index.php/ijphr/article/view/331
- Thank you for the suggestion. We have included the reference in the Discussion section (line 300).
Reviewer 3 Report
This is a good paper on a very important subject (tobacco quitting) and it is well written. However, the methodology needs some strengthening especially on the qualitative part. On data analysis, the authors mention the themes that emerged from the data analysis, however, information oh how the data was analysed is not indicated. There are many ways of analysing qualitative data, so the authors should indicate how they analysed data.
The limitation with qualitative studies is the issue of subjectivity. To ensure trustworthiness (rigorous) of the results, certain steps are taken during qualitative studies. The authors should explain how "rigour" was ensured on this study.
Different font sizes have been used in certain sections. It is recommended that the authors use uniform font sizes.
Author Response
We would like to thank all the reviewers for their comments and their time reviewing this manuscript. Responses to the reviewers are found below in bold. Edits to the manuscript have been tracked changed for ease of reviewing.
Reviewer #3
This is a good paper on a very important subject (tobacco quitting) and it is well written.
- Thank you for your comment.
- However, the methodology needs some strengthening especially on the qualitative part. On data analysis, the authors mention the themes that emerged from the data analysis, however, information oh how the data was analysed is not indicated. There are many ways of analysing qualitative data, so the authors should indicate how they analysed data.
- We have updated the Measures and Analyses section (lines 161-169) and the Procedure section (lines 111-113; 130; 135-136; 140-150) to clarify and provide more detail on the analysis of the qualitative data.
- The limitation with qualitative studies is the issue of subjectivity. To ensure trustworthiness (rigorous) of the results, certain steps are taken during qualitative studies. The authors should explain how "rigour" was ensured on this study.
- We appreciate your comment. We have included language and references in the Qualitative data section (lines 170-173) to highlight the guiding framework we used to ensure rigor of our qualitative research.
- Different font sizes have been used in certain sections. It is recommended that the authors use uniform font sizes.
- We have edited the formatting to ensure uniformity in font size throughout the manuscript.